# *Ehrlichia canis* Vaccine Development: Challenges and Advances

**DOI:** 10.3390/vetsci11120624

**Published:** 2024-12-05

**Authors:** Bruna Samara Alves-Ribeiro, Raiany Borges Duarte, Zara Mariana de Assis-Silva, Ana Paula Carvalho Gomes, Yasodaja Assis Silva, Lizandra Fernandes-Silva, Alice Caroline da Silva Rocha, Iago de Sá Moraes, Klaus Casaro Saturnino, Dirceu Guilherme de Souza Ramos, Isis Indaiara Gonçalves Granjeiro Taques, Ísis Assis Braga

**Affiliations:** 1Laboratory of Veterinary Parasitology and Clinical Analysis, Academic Unit of Agricultural Sciences, Federal University of Jataí, Jataí 75801-615, Goiás, Brazil; brunasamara@discente.ufj.edu.br (B.S.A.-R.); raiany.duarte@ufj.edu.br (R.B.D.); zaramariana@discente.ufj.edu.br (Z.M.d.A.-S.); anap_gomes@discente.ufj.edu.br (A.P.C.G.); lizandra_fernandes@discente.ufj.edu.br (L.F.-S.); iago.moraes@ufj.edu.br (I.d.S.M.); dguilherme@ufj.edu.br (D.G.d.S.R.); 2Laboratory of Veterinary Anatomical Pathology, Academic Unit of Agricultural Sciences, Federal University of Jataí, Jataí 75801-615, Goiás, Brazil; yasodajaassis@ufj.edu.br (Y.A.S.); klaus.sat@ufj.edu.br (K.C.S.); 3Academic Unit of Agricultural Sciences, Federal University of Jataí, Jataí 75801-615, Goiás, Brazil; alicerocha@discente.ufj.edu.br; 4College of Applied Social Sciences of the São Lourenço Valley, EDUVALE, Caiçara Street, 2.114, Jaciara 78820-000, Mato Grosso, Brazil; isis@eduvalesl.edu.br

**Keywords:** bioinformatics, canine monocytic ehrlichiosis, immunizing, immunoreactive proteins, tick-borne disease

## Abstract

Over the years, inactivated, live attenuated, and recombinant vaccines against *Ehrlichia canis* have been evaluated to mitigate the impact of canine monocytic ehrlichiosis. However, these results are not sufficient to produce an effective immunizing agent. New investigations can be conducted with the advancement of bioinformatics, immunoinformatics, and structural modeling tools, potentially ending a long-standing gap in research related to the development of vaccines against *E. canis*. Based on this technological advance, we are increasingly closer to developing a vaccine composed of multiple epitopes capable of eliciting robust humoral and cellular immune responses.

## 1. Introduction

Canine monocytic ehrlichiosis (CME) is an infectious disease caused by *Ehrlichia canis*, an obligate intracellular bacterium transmitted by ticks. It mainly affects domestic dogs but can also infect other domestic and wild animals. There have also been reports of humans parasitized by *E. canis* [1,2,3,4].

Despite the significant epidemiological impact of *E. canis* on public health and the veterinary field, a vaccine against this pathogen is not commercially available. The absence of lipopolysaccharides (LPS) in this bacterium invalidates strategies against glycoconjugates. It requires the identification of protein antigens, which has been a major hurdle for researchers, given the mechanisms of immunological subversion and the high antigenic variability of the species [5,6].

*In vitro* research and experimental needle inoculation, the latter a route used to infect dogs and murine models experimentally, are significant obstacles to vaccine development, as they do not reflect the reality of natural infection and, therefore, interfere with the practical assessment of the immune response [7]. However, after three decades of investigation, advances have been made in identifying the immunoreactive proteins of *E. canis*, such as p28, that acts as an outer membrane protein, the tandem repeat proteins (TRPs) TRP19, TRP36, TRP140, and ankyrin repeat protein (Ank200), some of which are targets for vaccine development [8].

Recent research has focused on developing immunogens for intracellular bacteria such as *E. chaffeensis*. Bioinformatics has streamlined the screening process, eliminating the need for *in vitro* and *in vivo* testing [9,10,11]. A team of researchers has successfully identified new hypothetical immunoreactive proteins in *E. canis* [12,13,14]. This discovery holds great promise for understanding the antigenic and immunogenic potential of these proteins.

The objective of this review is to present relevant aspects regarding the challenges and advances faced in the development of vaccines for *E. canis* and highlight perspectives for future investigations.

## 2. Canine Monocytic Ehrlichiosis

CME is an infectious disease caused by the obligate intracellular Gram-negative bacterium *E. canis* [2,3,4]. The ticks of the *Rhipicephalus sanguineus* complex represent the main vectors for transmission. Moreover, blood transfusion and vertical transmission are important routes of transmission [15,16]. The tick species have a heteroxenic cycle involving larval, nymph, and adult stages occurring in different hosts, making them susceptible to infection during blood feeding. However, only nymphs and adults can transmit the pathogen [17,18,19].

Despite its wide worldwide distribution, the prevalence of CME is higher in tropical and subtropical climate regions, where the *R. sanguineus* cycle is favored, especially during warmer periods of the year [15,20,21,22,23,24]. *R. sanguineus* has two well-defined lineages, temperate (*R. sanguineus* sensu stricto) and tropical (*R. sanguineus* sensu lato), distributed in the Americas and Western Europe and the Americas and South Africa, respectively [25,26,27,28]. Additionally, a new lineage was defined, showing genetic distinctness from *R. sanguineus* sensu lato between the Middle East and Eastern Europe regions [29].

Moraes-Filho et al. [30] and Sanches et al. [31] assign a greater vector competence to the tropical lineage compared to the temperate lineage. These studies provide essential data regarding the interaction of the *R. sanguineus* complex and *E. canis*. However, since new lineages are being discovered, further research will be necessary to determine the vector competence of these ticks. The tropical and temperate lineages of *R. sanguineus* are not only genetically distinct but also show differing proteomic reactions to infection. This fact supports the idea that they represent separate species with varying abilities to transmit *E. canis*. These findings aid in identifying potential targets for tick control and the prevention of tick-borne diseases, while also adding further insights into the ongoing discussion about the taxonomic classification of *R. sanguineus* [31].

After *E. canis* inoculation into susceptible dogs, the incubation period is 8 to 20 days, followed by acute, subclinical (asymptomatic), and chronic phases. The CME is a multisystemic disease and presents a series of challenges. As clinical and laboratory findings overlap with variable duration and severity, distinguishing each clinical phase in naturally infected dogs is a complex and challenging task. Hematologic abnormalities and clinical signs such as thrombocytopenia, anemia, depression, anorexia, weight loss, fever, and bleeding mark the acute phase. During the subclinical phase, dogs may either clear the infection spontaneously or remain infected while appearing clinically healthy for months to years. Eventually, some dogs may progress to a severe chronic infection characterized by hypoplastic bone marrow, bleeding, and death [32,33]. The development of clinical manifestations is related to several factors, such as the immunogenic and virulence characteristics of the strain involved in the infection, as well as the immune status of the host.

Genetic variability of *E. canis* has been reported in some isolates from different geographic regions after analysis of the 16S rRNA gene. Through analyses based on the tandem repeat protein 36 (TRP36) genes, it was possible to observe the variability of amino acid sequences among *E. canis* strains and generate genogroups. The genotypes include American (USTRP36), Brazilian (BrTRP36), Costa Rican (CRTRP36), Taiwanese, and Cuban (CUBTRP36), with different distributions on a global scale (Figure 1) [3,8,34,35,36,37,38,39,40,41,42]. Research reveals differences in the form of disease presentation in relation to the genotypes. Taques et al. [43], Borges et al. [41], and Zorzo et al. [44] have highlighted that the Brazilian genotype may be related to the chronic phase of CME because it is the most prevalent and adapted in Brazil. In contrast, the American genotype appears to be more pathogenic, causing more inflammatory signs. The Costa Rican genotype was less adapted to the immune response of animals from Brazil. Navarrete et al. [37] described dogs infected with the Cuban genotype as having more significant hemorrhagic tendencies, suggesting the possibility of the emergence of a more virulent strain.

*Ehrlichia* infection in immunocompetent dogs has been associated with the development of a Th1 response with IFN-γ secretion by CD4 T cells and changes in the CD4/CD8 ratio of T lymphocytes in peripheral blood, which may play pathogenic and protective roles [19]. Increases in the CD8+ cytotoxic T cell population have been detected in lymphoid organs and blood in experimental and natural cases of CME [32,45]. However, the dissemination of the bacteria may be facilitated by several evasion mechanisms that are initiated at the time of blood feeding by the vector, which contains components in its saliva that act as immunomodulators, reducing the production of IL-9, IL-2, and IL-4 in Th1 cells stimulated by IFN-γ [46]

The absence of LPS in *E. canis* increases its survival in cellular vacuoles and its ability to inhibit phagolysosomal fusion in infected cells [47,48]. This inhibition results from the suppression of proteins that form complex juxtaposing vesicular membranes that hinder fusion and reduce IFN-γ levels, which is inversely proportional to the frequency of *Ehrlichia*-infected cells [49].

Moreover, bacteria reduce the expression of the primary histocompatibility complex type I (MHC-I) and contribute to the maintenance of intracellular survival. During infection, there is a reduced expression of MHC-II, which is essential for the maturation of CD4+ T lymphocytes [32]. After 15 days of infection with *E. canis*, IgM immunoglobulins are detected in the bloodstream, marking the acute phase of the infection, while initial IgG levels are relatively low. As the infection progresses, predominantly comprising the IgG2 subclass, IgG titers increase notably, indicating the convalescent phase [50].

Tetracyclines continue to be the treatment of choice for ehrlichiosis in small animals, mainly doxycycline, but indiscriminate use has increased antimicrobial resistance [51]. Minocycline and oxytetracycline also act against these bacteria and are recommended when doxycycline is not available, refractory, or poorly tolerated [52]. Because of the many adverse effects of doxycycline treatment, alternatives for synergistic antimicrobial therapy of allopathic agents with natural plant actives are being studied [53]. Rosário et al. [52] showed that an aqueous fraction of *Ageratum conyzoides* associated with low doses of doxycycline has a promising, *in vitro*, anti-*Ehrlichia* action, being able to eliminate or suppress *E. canis*. Nonetheless, refractoriness and recrudescence of the bacteria have been reported after treatment [54,55].

Another challenging factor is ineffective tick control due to human failures and parasitic resistance mechanisms developed by vectors [56]. Sánchez-Pérez et al. [57] assumed, through mathematical models, a significant increase in *R. sanguineus* population in tropical and subtropical regions between 2050 and 2070 because of the climate changes currently being experienced, which will increase the suitable habitats for this tick’s survival.

## 3. Challenges for Developing Vaccine for *E. canis*

The challenges of formulating vaccines for CME are related to the complexity of the immunopathogenesis and the significant antigenic variability of *E. canis*, which make it difficult to select specific epitopes for effector cells. Since it is a tick-borne disease, the organic interactions required for the maintenance of the bacterium in the vector are also considered a challenge for disease control and the development of an effective vaccine. Differences in the microbiome of distinct *R. sanguineus* lineages have been described and suggest a direct relationship with the vector’s competence [58]. In addition, the study of salivary proteins expressed by the tick during blood feeding has proven to be fundamental for the vector’s development and effectiveness [59]. Additionally, the difficulty of maintaining *E. canis* culture, the use of *in vitro* techniques, and the routes and models used for experimental testing are limiting factors.

The host immune response effective to combat infection by bacteria of the *Ehrlichia* genus occurs through antigen presentation by conventional dendritic cells (CDC), followed by an acquired cellular immune response with activation of CD4+ cells in Th1 associated with humoral (Th2) responses [32,49]. The potential of vaccines targeting antigens for dendritic cells is a promising approach that may be effective against obligate intracellular bacteria. However, most research on the development of an immunogen for *E. canis* demonstrated the predominant triggering of humoral responses.

With the advancement of proteomics, several immunogenic proteins have been described and characterized as essential for the maintenance of infection. Outer membrane proteins (OMPs; P30/P28) [60], ankyrin repeat proteins (Anks) [61], and a group of proteins with amino acid repeat sequences (TRPs) have been identified in different species of *Ehrlichia* [50,62,63]. Luo et al. [64] reported that TRPs are involved in essential functions for the entry of *Ehrlichia* spp. into the host cell (phagocytosis, cytoskeletal reorganization, and intracellular transport) and replication mechanisms (cell signaling, metabolism, post-translational modification, and transcriptional regulation), in addition to interfering with the exit of *Ehrlichia* spp. in the infected cell (apoptosis and exocytosis). Bui et al. [65] reported that some TRPs are substrates of the type 1 secretion system (T1SS), which plays an important role in maintaining bacterial infection by secreting proteins that interact with a diverse network of targets in the host associated with essential cellular processes [66]. McBride et al. [50] highlight the importance of these proteins in immunodiagnosis and consider them essential targets for vaccine development [67].

Although TRP 19 is conserved among different isolates of *E. canis* and its ability to trigger antibody production has been verified, further studies need to be carried out on its potential to activate Th1 cells [62,63,67]. Similarly, another immunodominant protein, TRP36, has been studied for its immunogenic characteristics; however, because of the wide antigenic variation observed among *E. canis* isolates, its use as a vaccine antigen is limited. Aguiar et al. [34] described an *E. canis* isolate, #Cba16, that presented a unique combination of the American TR region and the N region of the Brazilian genotype. This finding suggests the possibility of genetic recombination owing to co-infection. It reinforces the hypothesis established by Doyle et al. [67], who stated that TRP36 is under a high level of selective pressure. Thus, it is believed that the effective immune response of a vaccine may be altered owing to differences in the pattern of the immune response in dogs exposed to different genotypes of *E. canis* and co-infection between them. In countries where there is a higher prevalence rate of CME and multiple reactivities, like Brazil, it is more challenging to develop an effective vaccine [40].

Another critical aspect to consider is the immune response produced by experimental needle inoculation, the route used to experimentally infect dogs and mice, compared to natural infection by tick transmission [68,69,70]. Patel et al. [70] reported that some immunoreactive proteins (TRPs) are highly expressed genes in mammalian cells. Nonetheless, most newly discovered proteins demonstrate higher expression levels in tick cells, suggesting they may be necessary for tick infection and transmission. The authors suggest that differences in host expression also illustrate how studies using only *Ehrlichia* produced in mammalian cells could be biased. However, they discuss the possibility of strongly immunoreactive proteins, highly expressed in ticks, likely to be important targets for transmission-blocking vaccines.

## 4. History of Vaccine Production for *E. canis*

Research on *E. canis* began in 1970, with a focus on the characterization of its pathogenesis, clinical description, and immunological response after infection. This research is fundamental for the development of vaccines against ehrlichiosis [71,72,73,74,75]. Notably, Ristic and Holland [75] played a pivotal role as pioneers in the evaluation of an immunizer despite the limited publicity of their study.

In 1998, Breitschwerdt [76] conducted a study on 16 dogs experimentally infected with *E. canis* to verify the efficacy of doxycycline in the treatment of CME. In addition to the therapeutic results, they provide the first insights into the essential implications for the development of a vaccine after considering the roles of innate and adaptive immune responses. Mahan et al. [77] investigated the response of German Shepherds after immunization with an inactivated *E. canis* combined with the Quil A adjuvant. A bacteremia-suppressive effect was observed after the challenge. However, Western blot analysis revealed a short-lived response, with a drop in antibody titer, exposing the animals to reinfection.

Or et al. [78] inoculated two dogs intravenously with a suspension of attenuated *E. canis* (Israeli strain) through multiple passages in culture media. The analysis consisted of verifying the transmission to naïve *R. sanguineus* and challenging the dogs after 119 days with a blood sample from a known infected dog. Nevertheless, the transmission of bacteria to ticks has not yet been verified. The immunized dogs developed bacteremia on day seven after the challenge, and one of them showed mild petechiae and splenomegaly. However, no hematological changes were observed in either dog. This study suggests the possibility of using *E. canis* attenuated by multiple passages in cell culture as a vaccine candidate. In another study using the same methodology, an additional study was conducted on 12 dogs, reinforcing the results described previously [79], but the immunity period was not evaluated in either study.

The use of primitive inactivated and live attenuated vaccines has led to undesirable effects and has not met expectations, and improvements in research using modern techniques have become evident. Advancements in reverse vaccinology have allowed for the identification of antigenic proteins, and the idealization of a peptide-based vaccine design has been employed [7,8,80,81]. In this context, research using murine models has been initiated to evaluate the dynamics of the immune response stimulated by synthetic peptides. The recombinant protein p29 from *E. muris* and the OMPs from *E. chaffeensis* p28 were the first targets of these investigations, and a significant reduction in bacterial load was observed, as well as the induction of a protective immune response in mice mediated by antibodies and T cells [82,83].

The identification of immunoreactive proteins of *E. canis*, followed by evidence of significant genetic and antigenic differences in antibody epitopes between the same *E. canis* strain [8,63], has stagnated the search for immunogens because of efforts to understand the immunopathogenic mechanisms of these synthesized proteins in natural infection. Moreover, the p19 protein of *E. canis* is the most conserved among these strains [36,84].

In a study, the *in vitro* neutralization capacity of hyperimmune serum synthesized from GP19 and produced in rabbits was evaluated, revealing promising data [85]. Subsequently, a prototype of a recombinant *E. canis* vaccine (rGP19) was tested in mice. The vaccinated group showed significantly higher mean antibody levels than the control group and a lower ehrlichial load in the blood. The authors suggested that immunogens effectively stimulate the immune response, enabling the organism to eliminate *E. canis* by stimulating CD4+ T cells, which produce IFN-γ, in addition to the production of antibodies [86].

The preliminary production of a vaccine against *E. canis* is challenging because of its low immunogenicity, difficulty in long-term protection, genetic variability, and undesirable effects. Since the SARS-CoV-2 pandemic, researchers have developed new technologies that combine immunoinformatics and cell-free protein expression, directly leading to the resumption of targeted research for immunoreactive antigens in the development of vaccines against *E. canis* [12]. Luo et al. [13,14] identified 18 immunoreactive proteins of *E. canis* (TRP) using bioinformatics. They hypothesized that some proteins had conformational epitopes and were type I secreted effectors (T1SS), highlighting them as future candidates for vaccine antigens.

Recently, Patel et al., [70] evaluated seven of these newly discovered immunodominant proteins and found that, except for one, all reacted early, between 21 and 28 days post-infection, according to an enzyme-linked immunosorbent assay. Furthermore, it was found that some of these proteins were highly conserved among *E. canis* strains and that the peptide E.caj_0126 R2 reacted with sera from positive dogs from North and South America (USA, Colombia and Brazil), demonstrating conserved epitopes among the geographically dispersed *E. canis* strains.

## 5. Prospects for Vaccine Development Against *E. canis*

After decades of research, we may be able to develop a vaccine against CME. Several studies have focused on identifying immunoreactive antigens of *E. canis* combined with reverse vaccinology and improving bioinformatics and structural modeling, which are essential for the development of vaccines against intracellular pathogens.

Bioinformatics is fundamental for identifying essential protein targets and non-host homologs in the pathogen proteome that can be used as potential vaccine candidate targets [9,87]. These proteins are generally involved in metabolic pathways critical for bacterial infectivity. Luo et al. [12,14] identified previously undiscovered hypothetical immunoreactive proteins in *E. canis* and provided additional options for immunogenic antigens. This discovery underscores the need for further studies to determine the T cell epitopes, secretion mechanisms, and functions of these proteins in *Ehrlichia* pathobiology and immunity. More recently, a linear antibody epitope termed E.caj_0126R2 has shown promise for future investigations, as it has been shown to be conserved among the different strains of *E. canis* distributed in the Americas [70].

The next crucial step is evaluation through immunoinformatics, focusing on targets that are immunodominant and effectors of the T1SS [14], as well as the proteins identified as TRP19, TRP36, TRP140, and Ank200 [68,80,86], to predict immune responses, particularly the interaction between epitopes and the host immune system, using tools that predict the antigenicity, allergenicity, and druggability of epitopes [88]. Research seeking to identify and evaluate pathogen epitopes unrelated to CME, transmitted by ticks, has also been conducted, with promising results [9,11].

Structural modeling is an essential component of computational vaccine design. Three-dimensional models of pathogenic proteins allow for the prediction of which protein conformations effectively trigger an immune response in a given host [88,89]. Computational tools, molecular docking simulations, and molecular dynamics are frequently employed to model the binding of epitopes to immune receptors and predict the stability and efficacy of potential vaccine candidates [90]. A multi-epitope vaccine against *E. chaffeensis* was designed, and immunological simulation analysis showed strong interactions with toll-like receptors and acceptable immunoreactivity, which this induced high levels of cytokine (IL-2 and IFN-γ), B cells, and T cell populations [11].

Conformational or mixed antigenic epitopes can be identified through *in silico* experimental models. They can also be expressed by the MHC-II of cDC that activates Th1 and Th17 cell receptors. This activation is crucial as it leads to the intense stimulation of the CD4+ T cell immune response and, consequently, the secretion of IFN-γ. Notably, CD4+ helper T cells play a pivotal role in driving the development of B cells, which differentiate into antibody-secreting plasma cells, memory cells, and long-lived plasma cells (LLPCs) that provide long-term and sustained antibody production. Epitopes for Th2 cell receptors are also desired, as a vaccine that produces a robust humoral, LLPC-stimulated, and T cell-mediated immune response will have a better chance of providing protection [6]. A strong cytotoxic cellular response to antigens is desirable. Although Th1 cells discretely stimulate the CD8 + T cell response, additional stimulation may be essential. The peptide presentation is required in a manner restricted to MHC-I, usually induced by antigen-presenting cells that secrete appropriate cytokines and co-stimulate cDCs to induce T cell differentiation [91] (Figure 2). Therefore, subunit vaccines against *E. canis* would require new formulations of adjuvants specialized in cross-antigen presentation [92].

After identifying the epitopes and understanding the essential characteristics of a robust immune response against *E. canis*, the next step is to define the correlates of protection (CoPs) for licensing an effective vaccine. This phase depends on humoral responses, well-defined CoPs, and cell-mediated responses that are predominantly tissue-resident memory T cells (TRM), as predicted by Schaik et al. [6].

After the *in silico* approach, prototypes of vaccines structured with multiple *E. canis* epitopes should be tested *in vivo* in the preferred host or in experimental models that mimic natural infections as much as possible. Budachetri et al. [93,94] investigated the vaccine potential of proteins synthesized by *E. chaffeensis*. They found a high production of antibodies and IFN-γ in immunized dogs, as well as rapid elimination of the bacteria. Vaccines against CME using immunoreactive proteins comprise two different delivery platforms: protein subunits and mRNA vaccines. The mRNA vaccines are more accessible and have a lower risk of adverse reactions [95].

Live attenuated vaccines (LAV) have been less explored in CME because of the difficulty in maintaining bacteria in cell culture. This methodology has been investigated using new attenuation approaches for the production of vaccines against another *Ehrlichia* species [96,97,98]. In one study, dogs were immunized with an attenuated mutant strain of *E. chaffeensis,* which produced antibodies and activated CD4+ T cells, consequently inducing protection for up to 12 months [98]. Similarly to LAV, DNA vaccines against *E. canis* have not been investigated. However, preliminary results indicate positive effects on the protection of *E. ruminantium* in animals [99,100].

Another prominent scenario involves strategies to control *E. canis* within the tick. Ferrolho and Dias have been working on studies highlighting potential targets, such as silencing ferritin and modulating the folate pathway to disrupt the pathogen’s replication in the vector and impair its survival [59,101].

## 6. Conclusions

The new methodologies and machine-learning algorithms developed during the SARS-CoV-2 pandemic have stimulated curiosity among researchers, renewing expectations for the formulation of a vaccine against *E. canis*. In just a few years, we have accelerated the process of discovering potential antigens. The subsequent phases of the investigation are set to proceed rapidly and safely, aided by bioinformatics, immunoinformatics, structural modeling, and *in vivo* validation experiments.

The challenges remain, as *E. canis* is an intracellular bacterium with high genetic variability and poorly understood evasion mechanisms. However, we are close to developing a vaccine against CME. From the authors’ perspective, the vaccine should be structurally composed of multiple conformational and linear epitopes. In addition to activating B cells, these epitopes will have a strong binding affinity to CD4+ and CD8+ T cells, promoting the production of cytokines, such as IFN-γ, that are essential for pathogen destruction. Additionally, control strategies aimed at disrupting the replication of *E. canis* in the vector would increase the chances of success in preventing CME.

## Figures and Tables

**Figure 1 vetsci-11-00624-f001:**
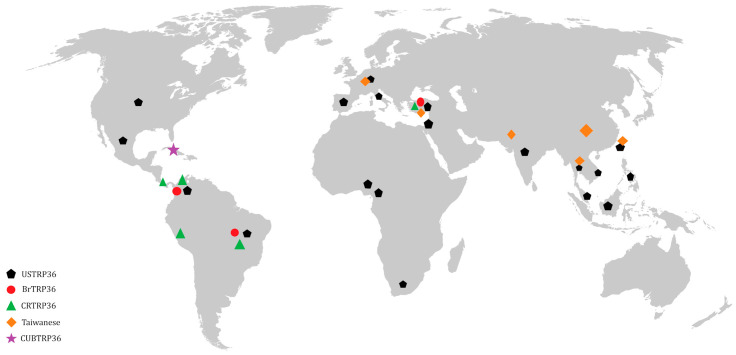
Geographical distribution of *Ehrlichia canis* genogroups based on the Tandem Repeat Proteins 36 (TRP36) genes reported in the literature until 2024 as genotypes: American (USTRP36), Brazilian (BrTRP36), Costa Rican (CRTRP36), Taiwanese, and Cuban (CUBTRP36).

**Figure 2 vetsci-11-00624-f002:**
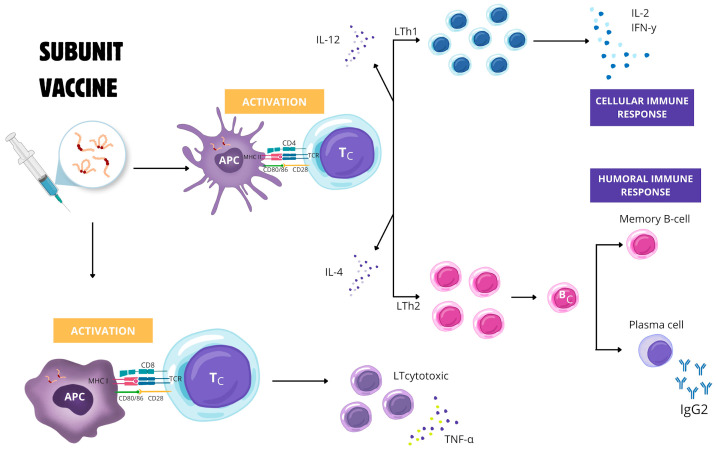
The immune response is stimulated by a vaccine antigen using a subunit vaccine with multiple epitopes (linear and/or conformational) of *Ehrlichia canis*. The presentation of cross-antigens to antigen-presenting cells (APC) is expressed by the major histocompatibility complex (MHC) type II and I processes. The molecules expressed by MHC-II activate Th1 cell receptors that stimulate the immune response of CD4+ T cells and, consequently, secretion of interferon-γ (IFN-γ) and interleukin-2 (IL-2). In contrast, activated Th2 cell receptors induce the development of B cells that differentiate into memory cells and antibody-secreting plasma cells. The molecules expressed by MHC-I induce the response of CD8+ T cells, triggering a cytotoxic response to antigens through the production of tumor necrosis factor-α (TNF-α).

## Data Availability

Not applicable.

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
