# Peer review of "Ehrlichia canis Vaccine Development: Challenges and Advances"

_vetsci, 2024, doi:10.3390/vetsci11120624_

Round 1
Reviewer 1 Report
Comments and Suggestions for Authors
Given the nature that this manuscript is a review I felt as though it was comprehensive and covered many aspects of the field of vaccine development in canine ehrlichiosis.
THE REVIEW INTENDS TO COVER THE CHALLENGES ASSOCIATED WITH VACCINE DEVELOPMENT FOR CANINE EHRLICHIOSIS AND THE WORK THAT HAS BEEN COMPLETED IN THE FIELD THUS FAR, INCLUDING POSSIBLE CANDIDATES.
THE WORK IS VERY RELEVANT TO THE FIELD, SINCE VACCINE DEVELOPMENT IN MANY OF THE TICK BORNE DISEASES HAS BEEN MINIMALLY ADVANCED AND THIS REVIEW PROVIDES A COMPLETE SUMMARY OF PAST, CURRENT AND FUTURE SCIENCE.
MUCH OF THE REVIEWS IN THIS FIELD FOCUS ON CLINICAL MEDICINE, PATHOGENESIS, AND PATHOGEN BIOLOGY. VERY FEW MANUSCRIPTS HAVE FOCUSED ON VACCINE DEVELOPMENT AND POTENTIAL CANDIDATES.
DUE TO THE NATURE OF THE REVIEW PAPER, METHODOLOGY IS NOT ABLE TO BE REVIEWED. THE APPROACH TO THE MANUSCRIPT IS WELL CONCEIVED AND PRESENTED IN AN ORGANIZED FASHION.
THE AUTHORS HAVE DONE AN EXCELLENT JOB OF REVIEWING THE SCIENCE IN THE FIELD THUS FAR, THE POTENTIAL CANDIDATES BASED ON SURFACE PROTEIN EXPRESSION, AND THE TYPES OF VACCINES THAT COULD BE DEVELOPED IF A VACCINE CANDIDATE WERE DEVELOPED, THEY DID A THOROUGH JOB OF SUMMARIZING THE RESEARCH THAT HAS BEEN PUBLISHED AND HOW THE CANDIDATES COULD PROVE EFFICACIOUS OR INADEQUATE TO DEPLOY AS A CANINE VACCINE.
REFERENCES ARE APPROPRIATE AND COVER THE FIELD BOTH IN BREADTH AND DEPTH.
THERE ARE FEW TABLES/FIGURES BUT THEY ARE ALL APPROPRIATE AND ENHANCE THE MANUSCRIPT.
Author Response
Comments 1:
Given the nature that this manuscript is a review I felt as though it was comprehensive and covered many aspects of the field of vaccine development in canine ehrlichiosis.
THE REVIEW INTENDS TO COVER THE CHALLENGES ASSOCIATED WITH VACCINE DEVELOPMENT FOR CANINE EHRLICHIOSIS AND THE WORK THAT HAS BEEN COMPLETED IN THE FIELD THUS FAR, INCLUDING POSSIBLE CANDIDATES.
THE WORK IS VERY RELEVANT TO THE FIELD, SINCE VACCINE DEVELOPMENT IN MANY OF THE TICK BORNE DISEASES HAS BEEN MINIMALLY ADVANCED AND THIS REVIEW PROVIDES A COMPLETE SUMMARY OF PAST, CURRENT AND FUTURE SCIENCE.
MUCH OF THE REVIEWS IN THIS FIELD FOCUS ON CLINICAL MEDICINE, PATHOGENESIS, AND PATHOGEN BIOLOGY. VERY FEW MANUSCRIPTS HAVE FOCUSED ON VACCINE DEVELOPMENT AND POTENTIAL CANDIDATES.
DUE TO THE NATURE OF THE REVIEW PAPER, METHODOLOGY IS NOT ABLE TO BE REVIEWED. THE APPROACH TO THE MANUSCRIPT IS WELL CONCEIVED AND PRESENTED IN AN ORGANIZED FASHION.
THE AUTHORS HAVE DONE AN EXCELLENT JOB OF REVIEWING THE SCIENCE IN THE FIELD THUS FAR, THE POTENTIAL CANDIDATES BASED ON SURFACE PROTEIN EXPRESSION, AND THE TYPES OF VACCINES THAT COULD BE DEVELOPED IF A VACCINE CANDIDATE WERE DEVELOPED, THEY DID A THOROUGH JOB OF SUMMARIZING THE RESEARCH THAT HAS BEEN PUBLISHED AND HOW THE CANDIDATES COULD PROVE EFFICACIOUS OR INADEQUATE TO DEPLOY AS A CANINE VACCINE.
REFERENCES ARE APPROPRIATE AND COVER THE FIELD BOTH IN BREADTH AND DEPTH.
THERE ARE FEW TABLES/FIGURES BUT THEY ARE ALL APPROPRIATE AND ENHANCE THE MANUSCRIPT.
Response: We would like to thank the reviewer for evaluating our manuscript, as well as for their compliments and considerations. We have attempted to address all of the reviewers' concerns appropriately and believe that our paper has improved considerably.
Reviewer 2 Report
Comments and Suggestions for Authors
The paper "Ehrlichia canis Vaccines Development: Challenges and Advances" provides a good overview of research on vaccine development against E. canis. However, several aspects could be added to improve the quality of the text.
Althought the role of the host’s immune system is discussed, the role of the vector is isn’t consider. Ticks have significantly impact on vaccine effectiveness, and considering strategies targeting this vector could enrich vaccine development against E. canis
Moreover, in recent years, the number of studies regarding tick microbiota and its impact on pathogen transmission have increased. This is a relevant topic in the development of vaccines for tick-borne pathogens and tick-borne diseases and is entirely omitted in the article. Including this perspective would be valuable, as there are already publications specifically examining the relationship between E. canis and tick microbiota.
Additionally, in the fields of immunoinformatics and structural modeling, models incorporating these variables could be explored to identify more effective antigens for the protection and control of Ehrlichia.
Discussing and considering all these aspects in the text could broaden the article’s relevance and applicability
Author Response
-
The paper "Ehrlichia canis Vaccines Development: Challenges and Advances" provides a good overview of research on vaccine development against E. canis. However, several aspects could be added to improve the quality of the text.
Althought the role of the host’s immune system is discussed, the role of the vector is isn’t consider. Ticks have significantly impact on vaccine effectiveness, and considering strategies targeting this vector could enrich vaccine development against E. canis
Moreover, in recent years, the number of studies regarding tick microbiota and its impact on pathogen transmission have increased. This is a relevant topic in the development of vaccines for tick-borne pathogens and tick-borne diseases and is entirely omitted in the article. Including this perspective would be valuable, as there are already publications specifically examining the relationship between E. canis and tick microbiota.
Additionally, in the fields of immunoinformatics and structural modeling, models incorporating these variables could be explored to identify more effective antigens for the protection and control of Ehrlichia.
Discussing and considering all these aspects in the text could broaden the article’s relevance and applicability
Response= We thank the reviewer for the excellent insights. We recognize that we failed to address this aspect and have added the following paragraphs:
- Line 91 - 94: The tropical and temperate lineages of R. sanguineus are not only genetically distinct but also show differing proteomic reactions to infection. This fact supports the idea that they represent separate species with varying abilities to transmit E. canis.
- Line 170 - 176: Since it is a tick-borne disease, the organic interactions required for the maintenance of the bacterium in the vector are also considered a challenge for disease control and the development of an effective vaccine. Differences in the microbiome of distinct R. sanguineus lineages have been described and suggest a direct relationship with the vector's competence [58]. In addition, the study of salivary proteins expressed by the tick during blood feeding has proven to be fundamental for the vector's development and effectiveness [59].
-
Line 369 - 372: Another prominent scenario involves strategies to control E. canis within the tick. Ferrolho and Dias have been working on studies highlighting potential targets, such as silencing ferritin one and modulating the folate pathway to disrupt the pathogen’s replication in the vector and impair its survival [102, 59].
- Line 387 - 388: Additionally, control strategies aimed at disrupting the replication of E. canis in the vector would increase the chances of success in preventing CME.
Reviewer 3 Report
Comments and Suggestions for Authors
Page |
Line |
Correction |
Suggestion |
3 |
120 |
Ehrlichia canis |
Use italics |
4 and 6 |
173 244 and 258 |
Immunizer (is it a sinonimous of immunogen?) |
I suggest to use immunogen or immunognic protein, (Correct and try to use in a consistent manner along the text) |
6 |
244-245 |
You mention in this sentence: “Immune response stimulated by immunizers containing synthetic peptides“ |
Rewritte this sentence. Usually, synthetic peptides (SPs) are used as immunogens, therefore SPs could not be a part of another immunogen. They are immunogenic peptides per se. |
6 |
252 |
mechanism |
Mechanisms |
6 |
252 |
Synthesized protein |
Make cleare this concept, It is not cleare if the protein was synthesized in vitro (sinthetic) or in vivo (recombinant). |
6 |
253 |
preserved |
Check the use of this word out: I am not sure if tyhe word preserved is a sinonimous of conserved. |
6 |
254 |
Synthesized |
Synthesized? or produced by inoculation of GP19 in animals? |
6 |
258 |
immunizer |
Rewritte the whole sentence: The immunogen do not eliminate the parasites, it activates the immune response so that, the organism can eliminate the parasite. |
7 |
273 |
conserved |
sometimes the same term is written as: conservd and some others as preserved, I suggest to use only one of them for consistency. |
7 |
290 |
conserved |
As above |
8 |
237 and 238 |
Major Histocompatibility |
Should say: Major histocompatibility complex |
|
|
GENERAL COMMENTS |
|
|
|
The information contained in this paper is important, However, I strongly suggest to review the grammar and syntaxis by a native english speaker. |
|
|
|
The chapter of references should be reviewed, since most scientific names are not written with italics. The name of the journals need format, for consistency all Journal names needs to be written with italics, Abreviations some times do not have dots and some others they do. |
|
|
|
I suggest that chapter 3 and 4 should be relocated. Chapter 3 should be 4 and backwards. |
|
|
The information contained in this paper is important, However, I strongly suggest to review the grammar and syntaxis by a native english speaker |
Author Response
- Line 120: Use italics in Ehrlichia canis / Response: done
-
Line 173, 244 and 258: I suggest to use immunogen or immunognic protein, (Correct and try to use in a consistent manner along the text / Response: done
- Line 244-245: Rewritte this sentence. Usually, synthetic peptides (SPs) are used as immunogens, therefore SPs could not be a part of another immunogen. They are immunogenic peptides per se / Response: done
- Line 252: Mechanisms / Make cleare this concept, It is not cleare if the protein was synthesized in vitro (sinthetic) or in vivo (recombinant)./ Response: done, and inserted in line 163: these "synthesized proteins in natural infection."
- Line 253: Check the use of this word out: I am not sure if tyhe word preserved is a sinonimous of conserved./ Response: yes, but the word was replaced by: conserved
- Line 254: Synthesized? or produced by inoculation of GP19 in animals?/ Response: line 266 replace "In a study, the in vitro neutralization capacity of hyperimmune serum synthesized from GP19 and produced in rabbits was evaluated"
- Line 258: Rewritte the whole sentence: The immunogen do not eliminate the parasites, it activates the immune response so that, the organism can eliminate the parasite. Response: Line 270 rewritten "The authors suggested that immunogens effectively stimulate the immune response, enabling the organism to eliminate E. canis by stimulating CD4+ T cells, which produce IFN-γ, in addition to the production of antibodies [86]."
- Lines 273 and 290: sometimes the same term is written as: conservd and some others as preserved, I suggest to use only one of them for consistency. / Response: done
- Lines 237 and 238: Should say: Major histocompatibility complex./ Response: done
- The information contained in this paper is important, However, I strongly suggest to review the grammar and syntaxis by a native english speaker./ Response: done
- The chapter of references should be reviewed, since most scientific names are not written with italics. The name of the journals need format, for consistency all Journal names needs to be written with italics, Abreviations some times do not have dots and some others they do. Response: done
- I suggest that chapter 3 and 4 should be relocated. Chapter 3 should be 4 and backwards./ Response: We appreciate the reviewer's suggestion; however, we have chosen to maintain the current order of topics, leaving the section 4 "History of Vaccine Production for E. canis" to convey continuity in the development process, showed in the section 5 'Prospects for Vaccine Development Against E. canis".
Reviewer 4 Report
Comments and Suggestions for Authors
This is a well-written and well structured review on the current state of strategies to develop an effective vaccine against Ehrlichia canis, an important tick-borne pathogen of dogs that is found worldwide.
The authors present a good overview of the disease and current challenges in preventing infections, then discuss the challenges of developing a vaccine against the disease. This is followed by a detailed review of the past, present and future prospects of vaccine development, outlining the various methods that have been attempted and the new generation of tools that are helping to predict promising vaccine candidates. The authors finish by defining the requirements and expectations for an effective vaccine and the steps that are needed to design and evaluate this candidate.
This is an excellent review of the current state of E. canis vaccination progress and is a good fit for this section of the journal. I only have minor comments to clarify some areas of the manuscript and highlight some errors for correction:
1. lines 75, 76: recommend correct the grammar here ".. main vectors for transmission.." and "...important routes of transmission."
2. line 84: both the tropical and temperate lineage of R. sanguineus have been detected in North America, so I suggest authors change "South America" to "the Americas" here.
3. line 92: clarify that this relates to infection in dogs. Could change the sentence to "After E. canis inoculation into susceptible dogs, ..."
4. line 183: clarify what is meant by "exit of the infected cell". Does it mean exit of Erhlichia bacteria from the infected cell, or destruction of the infected cell by apoptosis?
5. line 213: correct "History on..." to "History of..."
6. line 224: suggest changing "containing" to "combined with"
7. Figure 2: in top right of figure, IFNgamma is labelled INF instead of IFN
8. line 341: a word seem to be missing here - "multiple E. canis xxx" - should it be strains, antigen epitopes, ?
9. line 345: correct INF to IFN
Author Response
This is a well-written and well structured review on the current state of strategies to develop an effective vaccine against Ehrlichia canis, an important tick-borne pathogen of dogs that is found worldwide.
The authors present a good overview of the disease and current challenges in preventing infections, then discuss the challenges of developing a vaccine against the disease. This is followed by a detailed review of the past, present and future prospects of vaccine development, outlining the various methods that have been attempted and the new generation of tools that are helping to predict promising vaccine candidates. The authors finish by defining the requirements and expectations for an effective vaccine and the steps that are needed to design and evaluate this candidate./ Response: We would like to thank the reviewer for evaluating our manuscript, as well as for their compliments and considerations
This is an excellent review of the current state of E. canis vaccination progress and is a good fit for this section of the journal. I only have minor comments to clarify some areas of the manuscript and highlight some errors for correction:
- lines 75, 76: recommend correct the grammar here ".. main vectors for transmission.." and "...important routes of transmission."/ Response: done
- line 84: both the tropical and temperate lineage of R. sanguineus have been detected in North America, so I suggest authors change "South America" to "the Americas" here. / Response: done
- line 92: clarify that this relates to infection in dogs. Could change the sentence to "After E. canis inoculation into susceptible dogs, ..."/ Response: done
- line 183: clarify what is meant by "exit of the infected cell". Does it mean exit of Erhlichia bacteria from the infected cell, or destruction of the infected cell by apoptosis? / Response: done (line 194)
- line 213: correct "History on..." to "History of..." / Response: done (line 255)
- line 224: suggest changing "containing" to "combined with" / Response: done
- Figure 2: in top right of figure, IFNgamma is labelled INF instead of IFN / Response: done
- line 341: a word seem to be missing here - "multiple E. canis xxx" - should it be strains, antigen epitopes, ? / Response: add epitopes (line 355)
- line 345: correct INF to IFN / Response: done